# Border Handling for 2D Transpose Filter Structures on an FPGA

**Donald G. Bailey *** 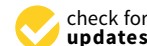 **and Anoop S. Ambikumar**

School of Engineering and Advanced Technology, Massey University, Palmerston North 4442, New Zealand;
A.Ambikumar@massey.ac.nz
* Correspondence: D.G.Bailey@massey.ac.nz

**Abstract:** It is sometimes desirable to implement filters using a transpose-form filter structure. However, managing image borders is generally considered more complex than it is with the more commonly used direct-form structure. This paper explores border handling for transpose-form filters, and proposes two novel mechanisms: transformation coalescing, and combination chain modification. For linear filters, coefficient coalescing can effectively exploit the digital signal processing blocks, resulting in the smallest resources requirements. Combination chain modification requires similar resources to direct-form border handling. It is demonstrated that the combination chain multiplexing can be split into two stages, consisting of a combination network followed by the transpose-form combination chain. The resulting transpose-form border handling networks are of similar complexity to the direct-form networks, enabling the transpose-form filter structure to be used where required. The transpose form is also significantly faster, being automatically pipelined by the filter structure. Of the border extension methods, zero-extension requires the least resources.

**Keywords:** stream processing; image borders; window filters; pipeline

---

## 1. Introduction

Image filtering is a common preprocessing operation in many image analysis applications. A local filter calculates the output for each pixel in an image as some function of the pixels within a local window in the input image. However, to produce an output for pixels on (or near) the image border, the input window extends past the edge of the input image. If such window pixels are not managed appropriately, the output pixels around the borders are invalid, and the effective image size shrinks. After a sequence of filters (especially if some of the filters are large), the effective image size can be substantially reduced, which is undesirable. Therefore, it is necessary to extend the input image through some form of extrapolation to provide suitable pixel values for the window pixels which extend past the borders of the input.

When processed using a field programmable gate array (FPGA), pipelined stream processing is the most common processing mode for implementing image filters [1]. Commonly, one pixel is processed per clock cycle (although this can be relatively easily generalized to two or more pixels per clock cycle [1]). To avoid memory bandwidth issues associated with reading all the input pixels for each window position, row buffers are typically used to cache pixels from previous image rows. Feeding the pixel stream through the window is equivalent to scanning the window through the image. Two commonly used window filter structures (ignoring image borders) are shown in Figure 1. The parallel structure shifts the window pixels in parallel with the row buffers, whereas the series structure uses the window pixels to extend the row buffers. When considering border handling, the parallel structure has the advantage that it decouples the vertical scanning (handled by the row buffers) from the horizontal

scanning. These filter structures are direct form, where the window is formed directly, followed by the filter function.

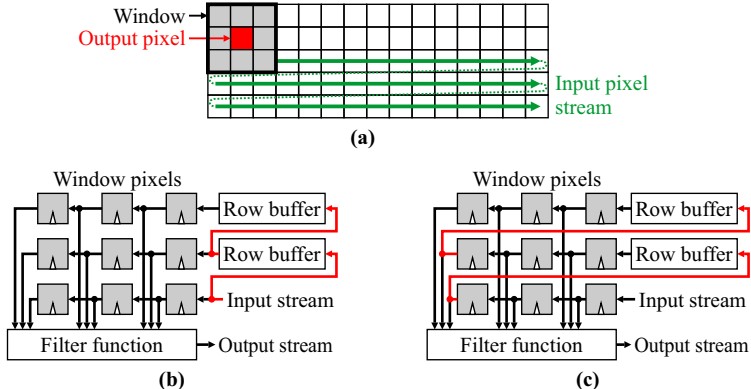

**(a)**

**(b)**          **(c)**

**Figure 1.** Window filters (without border handling). (**a**): window scanning; (**b**): parallel window structure; (**c**): series window structure; with the series and parallel connections highlighted in red.

The contribution of this paper is a systematic methodology for constructing border handling networks for 2D transpose-form filters. The remainder of this section reviews the transpose-form filter architecture, and commonly used border handling methods. Section 2 summarizes previously reported work on FPGA architectures for handling image borders. Two novel transpose-form border handling architectures are developed for 1D filters in Section 3, which are then extended to 2D filters in Section 4. Implementation of the architectures are discussed in Section 5 and compared experimentally in Section 6.

*1.1. Transpose-Form Filter Structures*

Although the filter function could be any function of the pixel values within the window, many useful filter functions can be represented as a transformation of each individual window pixel value, followed by a combination function which combines the transformed window pixels to produce a single output pixel value, as represented in Figure 2. Any filter which has an associative combination function can be restructured from direct form into a transpose form by [2]:

1.  interchanging input and output nodes,
2.  reversing all the paths through the filter,
3.  replacing branch nodes (pick-off points) with combination functions, and
4.  replacing combination functions with branch nodes.

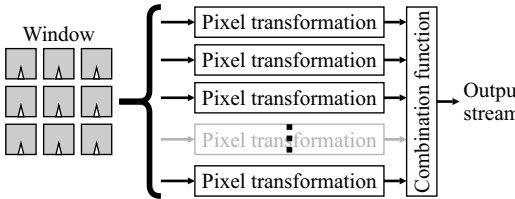

**Figure 2.** Filter function that can be arranged into transpose form.

Arguably, the most common filter function is the 2D finite impulse response (FIR) filter, where the output value is a weighted sum of the window pixels:

$$Q_{FIR}[x,y] = \sum_{i=-\frac{W-1}{2}}^{\frac{W-1}{2}} \sum_{j=-\frac{W-1}{2}}^{\frac{W-1}{2}} h_{i,j} I[x-i, y-j] \tag{1}$$

where $I[x, y]$ and $Q[x, y]$ are the input and output images respectively, with a $W \times W$ convolution kernel, $h_{i,j}$. The pixel transformation is multiplication of each window pixel by the corresponding filter coefficient, $h_{i,j}$, and the combination function is addition. Figure 3 shows the direct form $3 \times 3$ FIR filter restructured into its transpose form. Please note that reversing the computation order flips the filter coefficients relative to the direct form.

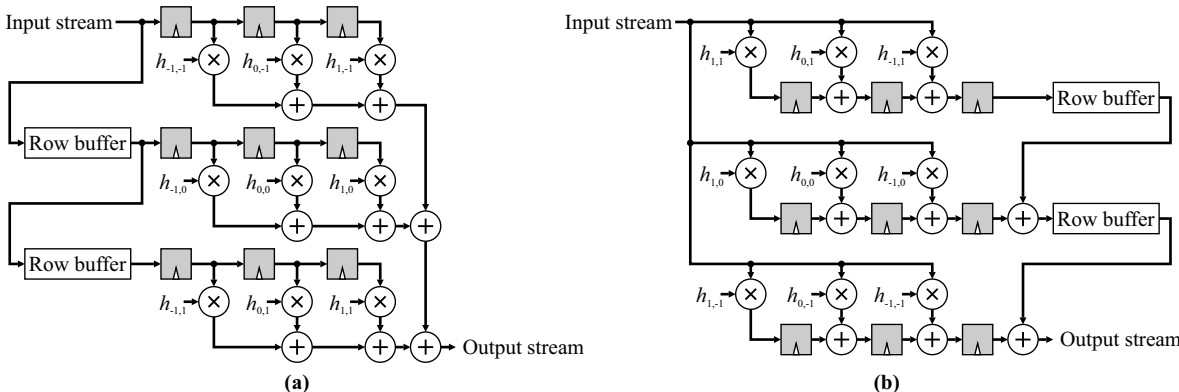

**Figure 3.** $3 \times 3$ FIR filter structure. (**a**): direct form; (**b**): transpose form.

Morphological erosion and dilation filters can also be arranged into transpose form. For example, greyscale dilation by a non-flat structuring element [3], $S[i, j]$, can be represented as:

$$Q_{dilation}[x, y] = \max_{i,j \in S} \left\{ I[x - i, y - j] + S[i, j] \right\}. \tag{2}$$

The pixel transformation is an offset of each window pixel value by the corresponding value of the structuring element, and the combination function is maximum. Similarly, for greyscale erosion [3]:

$$Q_{erosion}[x, y] = \min_{i,j \in S} \left\{ I[x - i, y - j] - S[-i, -j] \right\}. \tag{3}$$

The transpose form has the advantage that the combination function is automatically pipelined, by distributing the combination function over the window, as is clearly seen in Figure 3 for an FIR filter. Rather than calculating the output as the weighted sum of window pixels, the input pixels are weighted immediately, and are accumulated into the window position of the corresponding output pixel. Where there are common filter coefficients, for example with symmetric filters, the associated multiplications are in parallel, enabling the set of common multiplications to be replaced by a single multiplier [4]. In the direct form, this is equivalent to using the distributive property to factor out the common multiplications. When realized on an FPGA, the transpose form enables the multiplication, addition and register to be combined within a single digital signal processing (DSP) block, reducing the logic required for the implementation [5].

Similarly, with morphological filters (see Figure 4), the structuring element offsets (pixel transformations) are in parallel so any common offsets only need to be calculated once in transpose form. Again, the combination function is also pipelined automatically by the filter structure.

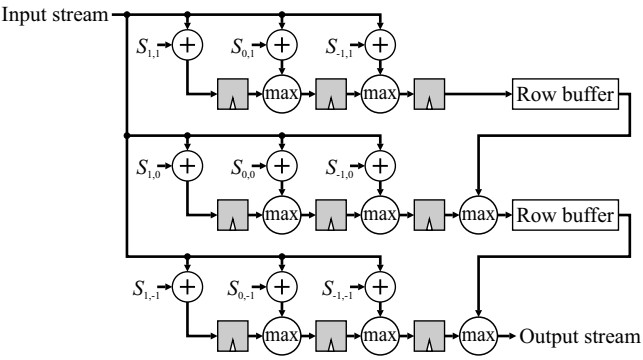

**Figure 4.** $3 \times 3$ greyscale dilation in transpose form.

When multiple filters are required in parallel (examples: Sobel filter, difference of Gaussians filter, sub-band filters, wavelet analysis), resources can be reduced by sharing the window structure (see left panel of Figure 5). However, when combining multiple filtered images together (examples: image fusion, high dynamic range imaging, wavelet synthesis), each filter must have its own window structure unless the transpose form is used (right panel of Figure 5). Of key importance is the ability to share the relatively expensive row buffers between all the parallel filters.

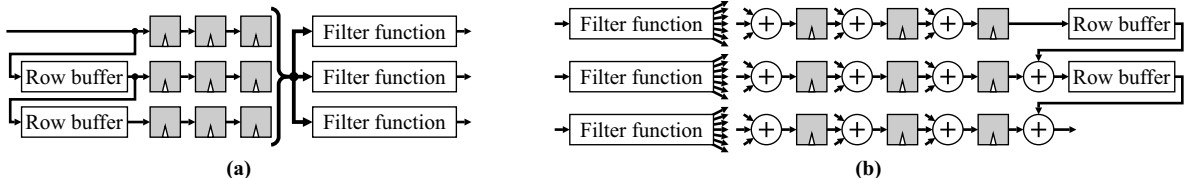

**Figure 5.** Shared windows for parallel filters (shown here for FIR filters). (**a**): direct form for distribution to multiple filters; (**b**): transpose form for collection from multiple filters.

One example where both direct form and transpose-form structures are used is in the parallel decomposition of morphological filters [6]. The flat structuring element is decomposed into parallel rectangular windows, which are each separable into row and column filters. Direct form is used to share resources (including row buffers) for the column filters, and transpose form is used to combine the results of the row filters, while sharing resources.

*1.2. Border Handling*

So far in this discussion, border handling issues have not been considered. The problem with simply processing the image is that pixels around the image border become corrupted because part of the window lies outside the image. In some machine vision applications, it may be possible to capture a larger image than required from the camera to allow for the reduction in image size as a result of subsequent filtering. Alternatively, it is important to ensure that the objects being imaged are kept sufficiently far from the image borders that important features of the objects are not corrupted by the processing. In such cases, no special processing is required around the image borders.

However, when filtering a video for enhancement, it is usually desirable for the output video to be the same size and format as the input. Some computer vision applications become less reliable when image borders are not processed appropriately. For example, Tan and Triggs [7] found that face detection (using difference of Gaussian filters) at the edges of the image was more reliable when the image was extended. Similarly, Jiang et al. [8] obtained improved saliency detection when the image was extended by reflecting edge super-pixels in the image border. For motion compensation with video coding, it is necessary to restrict the search for matching patches at the edges of the image. Sullivan

and Baker [9] made the observation that appropriate image extrapolation gave major improvement in coding performance.

Please note that any form of extrapolation of the image beyond the borders is estimating data that is not actually available. Consequently, if care is not taken, artefacts can be introduced into the image depending on image contents and filter being applied. Generally, better results are obtained by extrapolating the image before filtering rather than replacing the lost pixels by extrapolation after filtering [10].

In software, image borders are commonly managed by additional code to provide appropriate processing of border pixels. However, in hardware this can result in considerable extra logic solely for processing border pixels [10]. There are several different commonly used border handling methods, with the particular method selected based on the type of filter, and the expected image contents. These are enumerated here, with key methods illustrated in Figure 6.

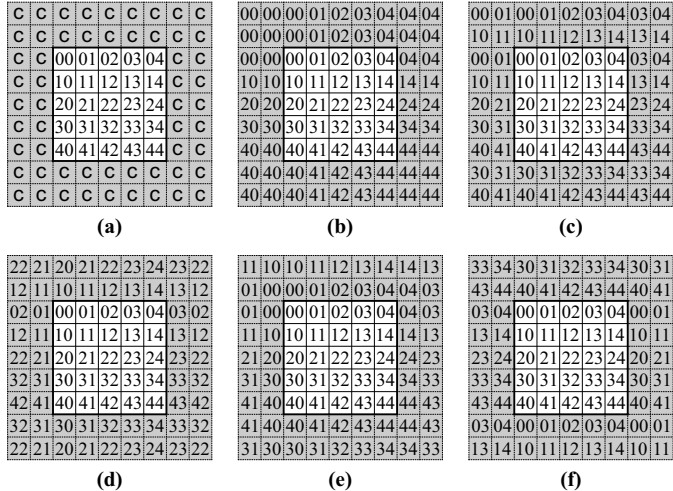

**Figure 6.** Image border extrapolation schemes (shaded pixels are extrapolated). (**a**): constant extension; (**b**): duplication; (**c**): two-phase duplication. (**d**): mirroring; (**e**): mirroring with duplication; (**f**): periodic extension.

1.  Do nothing [5,10,11]: This does not handle borders, and the effective size of the output image shrinks.
2.  Constant extension [5,10–12]: Pixels outside the image are assumed to be constant. Common constants are 0 (particularly with morphological processing) or the average value within the image.
3.  Duplication or clamping [5,10–12]: The nearest valid pixel within the image is used (zero order extrapolation).
4.  Two-phase duplication: Like duplication, but alternating the two outermost rows of the image. This is required, for example, when processing raw color images with a Bayer pattern or similar phased structure [13].
5.  Mirroring [5,10,12,14]: Pixels are mirrored about the outside row and column of pixels.
6.  Mirroring with duplication [5,10–12]: Pixels are mirrored about the image border such that the outside row and column are duplicated.
7.  Periodic extension (tiling) [10,11]: Extends the image by periodically tiling the image horizontally and vertically. This scheme is impractical for stream processing [10] because the whole image must be buffered to obtain the bottom row before processing the top row of the image. Opposite borders of an image usually have little correlation, resulting in artefacts around the borders of the output image.
8.  Modify the filter function [5,10–12]: An alternative to extending the input image is to explicitly modify the filter function to handle each scenario. When done naively, this can result in a large

amount of additional logic, making it less practical. It is generally more efficient to implicitly modify the filter function by modifying the formation of the window (using one of the other methods) and leaving the filter function unchanged [10].

With direct-form filters, window formation and the filter function are independent; managing borders simply involves selecting the required pixels to form the window that provides the pixel values to the filter function. The transpose form, however, is made more complicated because the window formation and filter function are more tightly integrated. The selection of pixels to form the windows generally requires multiplexing (depending on the position of the window relative to the border), and transforming structures containing multiplexers into transpose form is not trivial.

## 2. Prior Border Handling Architectures

The earliest work considering border handling architectures (other than zero-extension) was that of Chakrabarti [15], which proposed a routing network for mirroring (with duplication) the borders of 1D filters in the context of performing a discrete wavelet transform. The routing network was placed between the filter delay chain, and the filter function as shown in Figure 7. Although the original paper only considered a 4-pixel window and mirroring with duplication, it is easily generalized to wider windows and other border handling schemes.

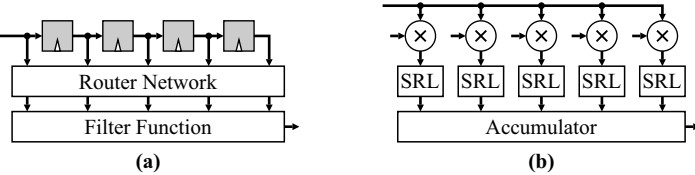

**Figure 7.** 1D border management techniques. (**a**): Chakrabarti [15]; (**b**): Benkrid et al. [14,16].

Benkrid et al. [16] used the FPGA logic elements as programmable shift register logic (SRL) to give variable delays for 1D FIR filters (see Figure 7). The accumulator was either an adder tree (direct form filter) or a pipelined adder chain (transpose form). Implementation details within [16] are sketchy; however more details are provided in a later paper [14]. The basic principle is that the shift register lengths are dynamically selected to route the required pixels to the accumulator to appropriately handle border conditions. Scheduling considerations increase the filter latency to the width of the window, a small but usually insignificant increase. However, this approach is only suitable for 1D filters, and cannot easily be generalized to 2D filter structures.

Bailey [10] considered direct-form 2D filters, using a parallel window structure to make the row and column processing independent. Two schemes were presented for row processing which exploited stream processing (where each pixel is only loaded once). Cached priming managed borders by routing pixels from where they were within the delay chain to where they were needed. The disadvantage of this method is that additional clock cycles are required between rows to flush the data from one row and load sufficient data to begin processing for the next row. The second method, overlapped priming and flushing, loaded the initial pixels from the next row into parallel registers while processing for the previous row was completed. At the end of the row, these were then transferred to appropriate locations within the window for the next row, avoiding the flushing and priming delays. These two methods are illustrated in Figure 8. For column processing, the row buffers were connected as a single chain, with multiplexing used to select the appropriate input for each window row. Column processing is therefore effectively the same as the routing network of [15].

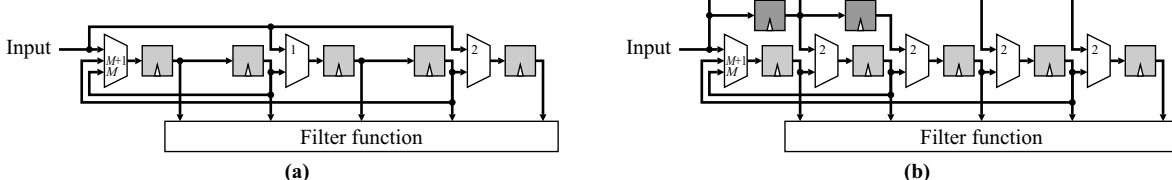

**Figure 8.** Row processing by Bailey [10] for mirroring without duplication. (**a**): cached priming; (**b**): overlapped priming and flushing.

Rafi and Din [11] also considered direct-form 2D filters, and replaced the parallel priming registers of [10] with multiplexers, effectively implementing the routing network of [15]. While this reduces the number of registers, the routing multiplexers are on the output of the window registers, potentially adding them to the critical path and reducing the clock frequency. Although this issue may be overcome by adding pipeline registers, this would negate the savings made.

Al-Dujaili and Fahmy's [5] focus was on optimizing speed by using DSP blocks for FIR filters. Although both direct and transpose filter forms were considered, for border management only the direct-form methods of [10] were implemented because of the complexities of border management with transpose filters.

## 3. Design for Transpose Filters

It is observed in Figure 6 that the border handling patterns are separable in the sense that extensions for the horizontal and vertical borders can be applied independently. This separability is independent of whether the underlying filter function is separable or not and enables the design of a 1-dimensional border handling mechanism and applying it to both the rows and columns within the window. This is made easier with the parallel window structure, where the row and column processing are separated. Referring to Figures 3 and 4, each row filter can be considered a pixel transformation for the vertical combination function performed via the row buffers. This section will therefore focus on structures for 1D transpose filtering, with the extension to 2D presented in Section 4.

For high speed processing, it is desirable to not introduce any additional clocking overheads between rows (and between frames). Therefore, when streaming one pixel per clock cycle, the last pixel in a row is immediately followed by the first pixel in the next row, and the last pixel in a frame is followed by the first pixel of the next frame. A complete $M \times N$ image is processed every $M \times N$ clock cycles, with the latency determined by the filter itself.

Consider the 1D linear FIR filter of width $W$ with filter coefficients $h_i$ given by

$$Q_{FIR}[x] = \sum_{i=-\frac{W-1}{2}}^{\frac{W-1}{2}} h_i I[x-i], \qquad 0 \leq x < M \tag{4}$$

where $I$ and $Q$ are the input and output images, respectively. Input pixels outside the range $0 \leq x < M$ are managed by the chosen border handling scheme. Figure 9 shows successive 5-pixel-wide row windows during the transition from one row to the next. Samples shaded grey represent invalid window pixels that must be replaced in the calculation through image border extension. The replacement is shown in bold, illustrating mirroring without duplication in Figure 9. These are linked to the corresponding source pixels with an arrow. The direction of the arrow is to the source pixel, because in the transpose form, the timing is governed by when a pixel is input. Additional processing is required therefore when the circled pixels are input.

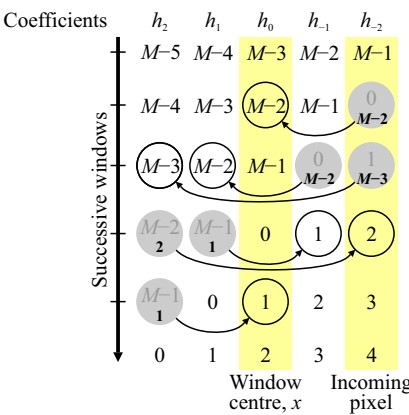

**Figure 9.** Border management for mirroring without duplication.

Two approaches to applying this additional processing are proposed. The first is to modify the filter function by coalescing the pixel transformations. The second applies the modification to the pipelined combination chain.

### 3.1. Transformation Coalescing

With border handling, some of the input pixel values appear at more than one position within the window and are therefore subjected to more than one-pixel transformation. For example, in an FIR filter, the input pixel is multiplied by more than one filter coefficient. Consider the window at $x = 0$ in Figure 9:

$$
\begin{aligned}
Q_{FIR}[0] &= h_{-2}I[2] + h_{-1}I[1] + h_0I[0] + h_1I[1] + h_2I[2] \\
&= h_0I[0] + (h_{-1} + h_1)I[1] + (h_{-2} + h_2)I[2] \\
&= h_0I[0] + h'_{-1}I[1] + h'_{-2}I[2].
\end{aligned}
\tag{5}
$$

The samples can be factored out, giving modified filter weights as combinations of the original filter weights, i.e.,

$$
\begin{aligned}
h'_{-1}[1] &= h_{-1} + h_1 \\
h'_{-2}[2] &= h_{-2} + h_2
\end{aligned}
\tag{6}
$$

These are labelled with positions 1 and 2 respectively, because these modified weights need to be applied to those corresponding input samples.

This pairing is applied for each arrow within Figure 9, with the label being given by the head of the arrow. The corresponding set of coalesced coefficients then become:

$$
\begin{aligned}
h'_2 \ \ [M-3] &= h_2 + h_{-2} \\
h'_1 \ \ [M-2] &= h_1 + h_{-1} \\
h'_0 \ \ [M-2] &= h_0 + h_{-2} \\
h'_0 \ \ [1] &= h_0 + h_2 \\
h'_{-1} \ \ [1] &= h_{-1} + h_1 \\
h'_{-2} \ \ [2] &= h_{-2} + h_2
\end{aligned}
\tag{7}
$$

It is also necessary to set the coefficients for the invalid samples (shaded grey in Figure 9) to 0 to prevent the invalid samples from being accumulated:

$$
\begin{aligned}
h'_2 \ [M-2] &= 0 \\
h'_2 \ [M-1] &= 0 \\
h'_1 \ [M-1] &= 0 \\
h'_{-1} \ [0] &= 0 \\
h'_{-2} \ [0] &= 0 \\
h'_{-2} \ [1] &= 0
\end{aligned}
\tag{8}
$$

The resulting coalesced coefficients can be formed using a multiplexer indexed by the sample number as demonstrated in Figure 10. Obviously, this can be simplified if the coefficients are constants because the sum of constants is also a constant.

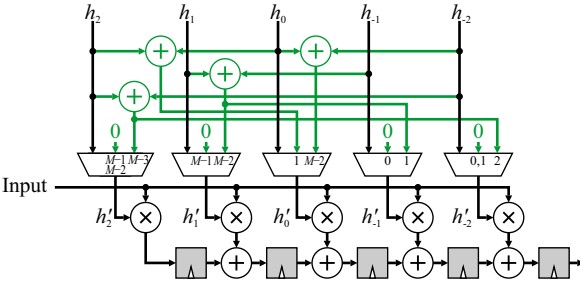

**Figure 10.** Coefficient coalescing for mirroring without duplication.

Please note that coefficient coalescing is only applicable to FIR filters. This technique may or may not be able to be adapted to other filter types depending on the filter function. For example, with a greyscale dilation filter, with structuring element offsets $S_i$

$$
Q_{dilation}[x] = \max_{i \in S} \{I[x - i] + S_i\}, \qquad 0 \le x < M
\tag{9}
$$

From Figure 9, the window position at $x = 0$ gives:

$$
\begin{aligned}
Q_{dilation}[0] &= \max \{I[2] + S_{-2},\ I[1] + S_{-1},\ I[0] + S_0,\ I[1] + S_1,\ I[2] + S_2\} \\
&= \max \{I[2] + \max(S_{-2}, S_2),\ I[1] + \max(S_{-1}, S_1),\ I[0] + S_0\}
\end{aligned}
\tag{10}
$$

with the corresponding set of coalesced offsets, $S'$, as:

$$
\begin{aligned}
S'_2 \ [M-3] &= \max(S_2, S_{-2}), & S'_{-2} \ [1] &= -\infty \\
S'_1 \ [M-2] &= \max(S_1, S_{-1}), & S'_{-1} \ [0] &= -\infty \\
S'_0 \ [M-2] &= \max(S_0, S_{-2}), & S'_{-2} \ [0] &= -\infty \\
S'_0 \ [1] &= \max(S_0, S_2), & S'_2 \ [M-1] &= -\infty \\
S'_{-1} \ [1] &= \max(S_{-1}, S_1), & S'_1 \ [M-1] &= -\infty \\
S'_{-2} \ [2] &= \max(S_{-2}, S_2), & S'_2 \ [M-2] &= -\infty
\end{aligned}
\tag{11}
$$

where the offsets of $-\infty$ correspond to the entries removed from the window (shaded grey in Figure 9). The resulting architecture, in Figure 11, has obvious parallels with coefficient coalescing for FIR filters in Figure 10.

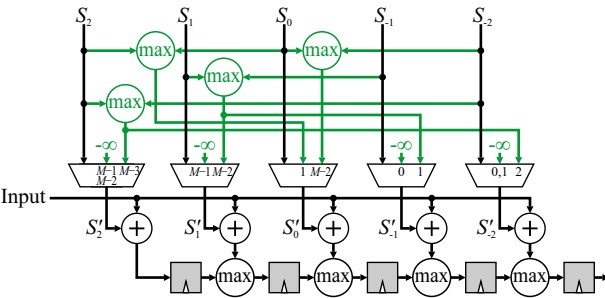

**Figure 11.** Transform coalescing for greyscale dilation using mirroring without duplication.

### 3.2. Combination Chain Modification

This next technique is applicable to all filters which are transposable. The pixel transformation is left unchanged, and the border handling is applied to the pipelined combination chain. This method works by combining the terms resulting from pixels outside the image into the appropriate place within the combination chain based on the source of the duplicated or mirrored pixel.

A $5 \times 5$ FIR filter using mirroring without duplication will be used to illustrate the method. Again, the parallel filter structure is used for separability, and the construction in Figure 9 is used to develop the modifications. Each arrow indicates where a transformed pixel needs to be combined into a different place within the output chain. Consider the window centered at $M - 2$. The window extension is $I[M - 2]$ (shown in grey) which is multiplied by $h_{-2}$. To arrive at the output at the correct time, the product $h_{-2}I[M - 2]$ must be added with the regular contribution from pixel $M - 2$ (shown by the circle), which is the center of the window. Since this combination is added only during clock cycle $M - 2$, a multiplexer is required. All 6 window extensions are shown in green in Figure 12.

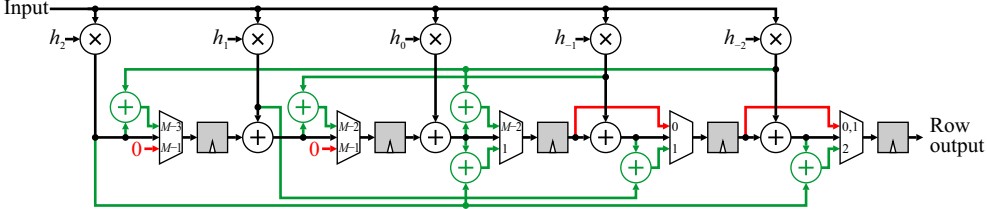

**Figure 12.** Row combination chain for mirroring without duplication.

It is also necessary to prevent the invalid pixels (grey circles in Figure 9) from being accumulated. On the right-hand part of the window, this may be achieved by bypassing the adder on those clock cycles. On the left-hand part of the window, it is easier to just clear the registers on the last pixel of a row. This may be accomplished either through multiplexing, or directly using a synchronous clear of the associated flip-flops. These modifications are shown in red in Figure 12.

Please note that the number of additional combination functions (additions in the case of FIR filters) and associated multiplexers is given by the number of shaded entries in Figure 9 and grows quadratically with the window size $W$ ($W$ is assumed odd here):

$$C_{1D} = \left(\frac{W - 1}{2}\right)\left(\frac{W + 1}{2}\right). \tag{12}$$

However, several of the combination terms are common, and this can be exploited by moving the multiplexers to the input of the combination chain. Where there are multiple combinations for a given stage, these will usually occur on different clock cycles. Moving the multiplexer to before the combination operation enables a single combination operation to be reused, as shown in Figure 13. This reduces the number of adders required to

$$C_{1D} \leq W, \tag{13}$$

which can result in a significant savings for larger windows. However, the complexity of the multiplexers increases with the size of the window as more terms must be multiplexed together before the combination for larger windows.

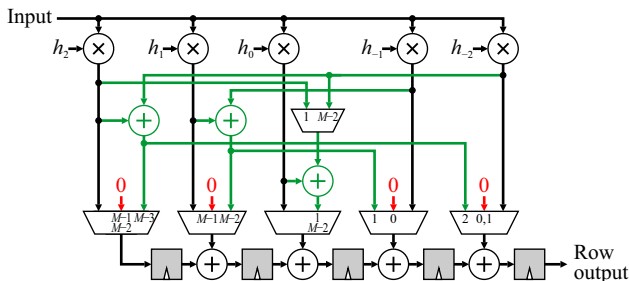

**Figure 13.** Optimization of row combination chain for mirroring without duplication.

### 3.3. Constant Extension

The above proposals manage the image border using a nearby window pixel. However, they do not work directly with constant extension.

The simplest is zero-extension. For FIR filters, when using coefficient coalescing, this can simply be achieved by setting the coefficients for invalid pixels to 0. For the modified combination chain, the extended combinations do not need to be added in (they are 0). It is only necessary to ensure that the invalid combinations are not added in, by bypassing the adders for the right-hand registers and clearing the left-hand registers. These approaches are shown in Figure 14. Note: they do not necessarily work directly with other filter types.

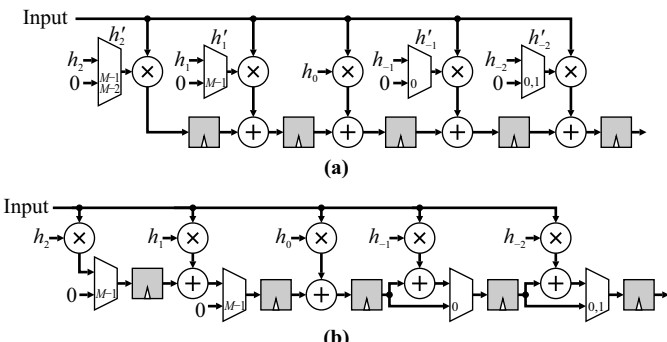

**Figure 14.** Zero-extension. (**a**): coefficient coalescing; (**b**): modified combination chain.

For constant extension, perhaps the simplest approach is to simply multiplex the input for those samples outside the image (shaded in grey in Figure 9). This is shown for row processing within an FIR filter in Figure 15.

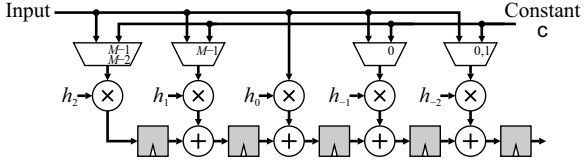

**Figure 15.** Constant extension.

### 4. 2D Filters

So far, all the examples that have been given apply to the 1D row filter components. Since border handling is separable (regardless of the separability of the underlying filter function) the same techniques can be applied for column handling. This is illustrated in Figure 16, where each row filter consists of a pixel transformation (multiplication by the FIR filter coefficient), combination network (for managing image borders, such as shown in Figure 13), followed by the combination chain. In 2D, pixel transformation consists of the set of row filters, the border managing combination network routes each row filter output into the appropriate tap of the combination chain, which now consists of row buffers rather than registers to combine data from different rows.

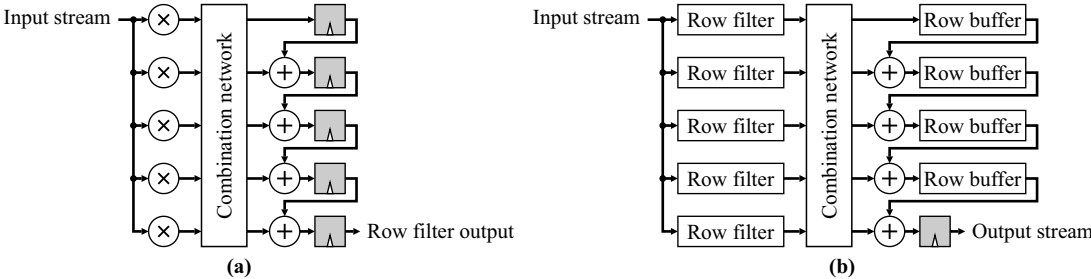

**Figure 16.** Extending 1D to 2D. (**a**): a 1D row FIR filter; (**b**): column combination processing for a 2D FIR filter.

If a 2D filter is not separable, then each row filter would be different. In transpose form, these row filters are in parallel, enabling reuse to be exploited (for example in the case of symmetry). The combination network manages the top and bottom image borders following the same principles as row border management. A specific example of the combination network for mirroring without duplication is illustrated in Figure 17. The combination network is identical to that shown in Figure 13 with the exception that the multiplexers are controlled by the row number rather than the column number.

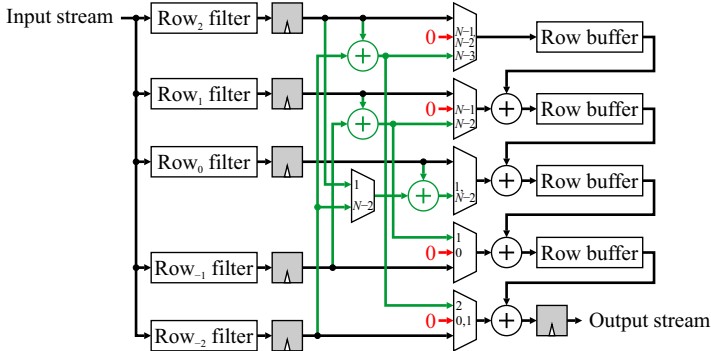

**Figure 17.** Column combination chain for mirroring without duplication.

Transform coalescing can also be extended to 2D. In the previous section, FIR filter coefficients were coalesced horizontally across the row. Since the border extension is separable, the same technique can be used to coalesce the resulting coefficients vertically. This is illustrated in Figure 18 where the coefficients $h'_{i,j}$ are formed from coalescing across row $j$. The same network is then applied vertically (down each column $i$), indexed by row number, with a second level of multiplexers selecting $h''_{i,j}$ from combinations of $h'_{i,j}$. Please note that 2D transform coalescing requires applying the coalescing to each column of the filter, rather than a single combination chain modification to the outputs of the row filters. The number of multiplexers is proportional to the number of pixels within the window, and the

complexity of the multiplexers (number of terms that must be multiplexed) will also grow with the window width.

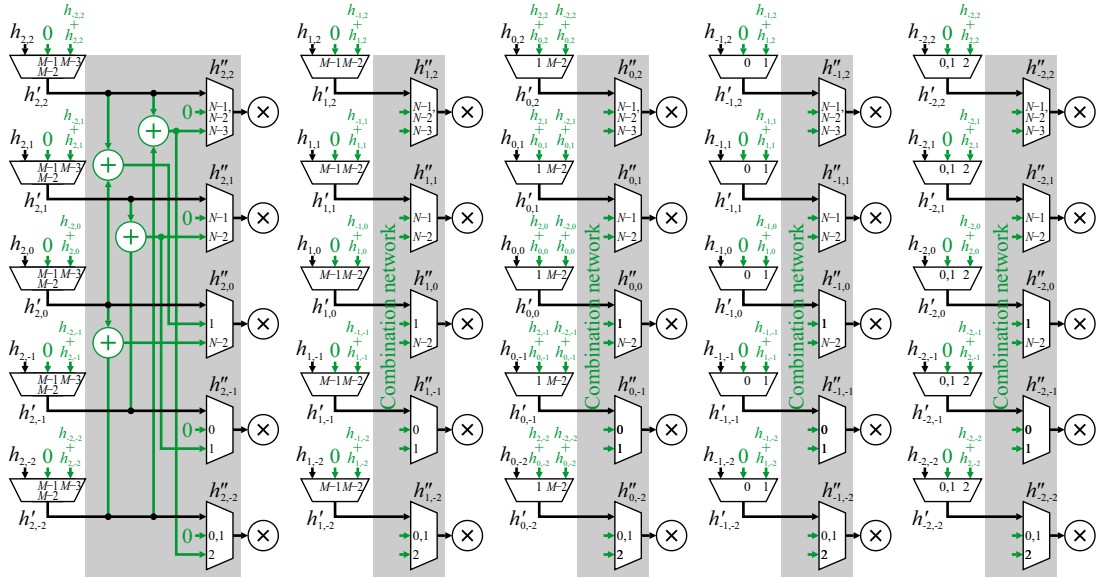

**Figure 18.** Coefficient coalescing by columns for 2D FIR filters (5 × 5 for mirroring without duplication).

The constant extension scheme shown in Figure 15 can similarly be adapted for 2D filtering by adding a second set of input multiplexers controlled by the row numbers.

*Control Circuitry*

In the proposed architectures, the multiplexers are controlled by the timing associated with the incoming pixel stream. This timing can easily be provided by a horizontal column ($x$) counter (modulo $M$) and a vertical row ($y$) counter (modulo $N$), which are reset by the corresponding stream synchronization signals. Rather than explicitly decode each of the required states, it is more efficient to decode the earliest state during a transition at an image border, and use a shift register to decode the subsequent states, as demonstrated in Figure 19. For the row counter and state shift register, providing a clock enable signal from the last column ($M - 1$) minimizes the additional circuitry required. (If the rows and columns are extended separately, then the clock enable can be shifted to a later tap to account for the latency of the row filters).

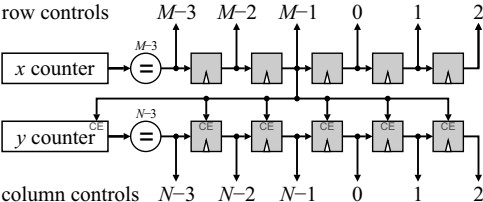

**Figure 19.** Control signal decoder.

## 5. Linear Filter Implementation Issues

One practical issue with implementing FIR filters is that the coefficients are usually scaled to enable integer arithmetic to be used. Where possible, the scale factor is a power of 2 (effectively representing the coefficients as binary fixed point), so the output of the filter can be obtained simply by bit-shifting (free in hardware). Rescaling is usually the last operation within the filter to minimize the precision loss through the adder chain. One implication of this is that the word length is larger on the output

of the multipliers. In particular, this affects the word length (hence memory size) of the row buffers, which for transpose filters must be sized according to the adder chain rather than the filter input. The buffer word length may be reduced by distributing the rescaling (as illustrated in Figure 20), with a partial rescaling of the outputs of the row filters, and the balance applied at the final filter output. The output can be rounded by adding in the scaled equivalent of 0.5 at the start of the chain.

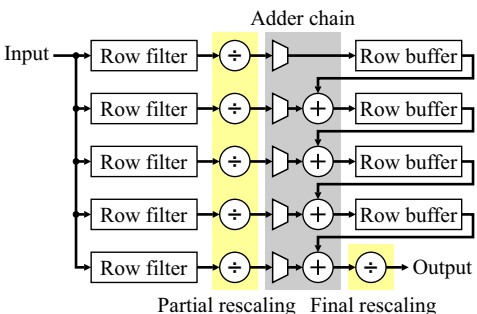

**Figure 20.** Distributed rescaling for transpose-form FIR filters.

If necessary for speed, coefficient coalescing can be pipelined, with a register placed between the multiplexer and multiplier. This would require adjusting the multiplexers to select the required coefficients 1 clock cycle earlier, which can easily be achieved. On an FPGA, this would generally use the flip-flop on the output of the multiplexer logic cell, which would otherwise be unused.

On modern FPGAs, the multiplication is realized within hard DSP blocks. Speed can be optimized, and additional resources minimized if the multiplication, addition, and following register can be implemented within a single DSP block [5]. Indeed, this is the case with the coefficient coalescing scheme, as seen in Figure 10. In fact, for Intel FPGAs, if the filter coefficients are constant and there are 8 or fewer combinations for each coalesced coefficient, then the coefficient multiplexing can also be implemented directly within the DSP block [17], although this requires direct instantiation of DSP primitives and cannot be inferred from the register transfer level (RTL) source code.

The disadvantage of coefficient coalescing is that it is no longer possible to reuse common multipliers within the filter. However, since the combination chain modification scheme moves the multiplexers to after the multipliers, such multiplier reuse is possible.

Modifying the combination chain prevents the implementation of both the multiplication and following addition within a single DSP block because the output of the multiplier is used in more than one place. However, for typical word sizes used for images, 2 or even 3 multipliers may be able to be realized by a single DSP block [17].

## 6. Experimental Comparison

To compare the different border management approaches, three experiments were performed. The first compares the cost of border management over doing nothing. For this, the transformation coalescing and combination chain modification techniques are compared for both FIR and morphological filters. For comparison, the transpose form is also compared against the more traditional direct-form filter structures. The second experiment compares the cost of different border extension methods for transpose-form filters. The third experiment investigates the scalability of border extension with window size.

All experiments were performed with symmetrical but non-separable square windows. Obviously, separable windows would significantly reduce resource requirements by implementing the 2D window as a cascade of two 1D windows. Non-separable filters require implementation of the full 2D window. Many filters are symmetrical, and resources can be reduced by exploiting symmetry where possible. However, border management will damage symmetry if care is not taken, for example with the transformation coalescing methods, and a symmetrical filter would enable this aspect to be explored.

The first two experiments used 5 × 5 filters, with the FIR filter coefficients and morphological filter structuring element offsets as shown in Figure 21. The FIR filter is a Gaussian filter, a frequently used filter, and although the Gaussian is technically separable, as a result of rounding, this filter is not separable, requiring implementation of a full two-dimensional window. The symmetry of both filters enables the number of computations to be reduced by a factor of approximately 2 when symmetry can be exploited. The test designs processed a 1024 × 768 × 8-bit image. For all FIR filter implementations, fixed-point arithmetic was used throughout, with the output pixel values obtained by rounding. Rounding was achieved by adding 0.5 at an appropriate point (usually at the start of the adder chains) and truncating the fraction bits of the final addition.

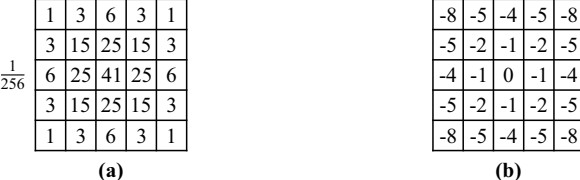

**Figure 21.** Coefficients for the sample filters used for testing in experiments 1 and 2. (**a**): FIR filter; (**b**): morphological filter structuring element offsets.

Designs were represented using VHDL, and synthesized using Quartus Prime Lite Edition 17.1.0 (Intel, Santa Clara, CA, USA), using "Balanced optimization (Normal flow)" targeting a Cyclone V FPGA (5CSEMA5F31C6) (Intel, Santa Clara, CA, USA) to give the resource requirements and the design speed. Modelsim-Intel FPGA Starter Edition 10.5b (Intel, Santa Clara, CA, USA) was used for design verification.

## 6.1. Experiment 1: Comparison of Approaches

The aim of this experiment was to compare the performance of the different architectural alternatives described in Sections 3 and 4 with both the FIR filter and the morphological filter. The synthesis results are summarized in Table 1.

**Table 1.** Resource requirements for the different approaches (adaptive lookup tables (ALUT), flip-flops (FF), 10 kbit memory blocks (M10K), DSP blocks (DSP), maximum clock frequency (Fmax) in MHz). Other than the cases with no border management (indicated by *), mirroring without duplication is used.

| Method | 5 × 5 FIR Filter | | | | | 5 × 5 Morphological Filter | | | |
|---|---|---|---|---|---|---|---|---|---|
| | ALUT | FF | M10K | DSP | Fmax | ALUT | FF | M10K | Fmax |
| Transpose form, no borders | | | | | | | | | |
| – DSP-based (*) | 164 | 100 | 7 | 9 | 175 | | | | |
| – Coefficient sharing (*) | 312 | 237 | 7 | 2 | 176 | 285 | 167 | 4 | 179 |
| Transformation coalescing | 296 | 131 | 7 | 15 | 120 | 688 | 251 | 4 | 184 |
| – Pipelined | 296 | 135 | 7 | 15 | 173 | 710 | 300 | 4 | 186 |
| Combination chain | 410 | 251 | 7 | 2 | 180 | 439 | 171 | 4 | 172 |
| – Partial rescaling 4/4 | 389 | 247 | 5 | 2 | 194 | | | | |
| – Partial rescaling 6/2 | 378 | 245 | 4 | 2 | 180 | | | | |
| Direct-form filter structures | | | | | | | | | |
| – No border handling (*) | 331 | 265 | 4 | 5 | 55 | 731 | 265 | 4 | 56 |
| – Multiplex network | 503 | 267 | 4 | 5 | 57 | 745 | 266 | 4 | 56 |
| – Adder/max tree | 380 | 267 | 4 | 2 | 93 | 580 | 267 | 4 | 64 |
| – Overlap prime and flush | 404 | 346 | 4 | 5 | 56 | 807 | 346 | 4 | 56 |
| – Adder/max tree | 317 | 346 | 4 | 2 | 105 | 500 | 346 | 4 | 68 |
| – Pipelined | 343 | 398 | 4 | 2 | 152 | 473 | 395 | 4 | 117 |

As a baseline, the transpose form with no border management was implemented in two ways for the FIR filter. The first, exploited the use of DSP blocks on the FPGA, since these are specifically targeted for multiply and accumulate operations. The second exploited the fact that many of the filter coefficients were used multiple times, with only 6 multipliers (one for each unique coefficient) shared for the complete window.

As expected, the DSP-based design used the fewest resources. Only 3 rows of the filter were required, because of symmetry, and each row used 3 DSP blocks because two multiply-add and associated output registers were able to be packed per DSP block. Coefficient sharing reduced the number of hardware multipliers to 2, since multiplication by 1, 3, 6, and 15 were optimized to additions. The adder chains were no longer implemented using DSP blocks, giving an increase in both logic and flip-flops. The four 16-bit wide row buffers (64 bits total) were packed into 7 M10K memory blocks.

For the morphological filter, coefficient sharing reduced the number of offsets required to 5. Any offset pixel values that went below 0 were clipped at 0. The four 8-bit wide row buffers (32 bits total) required 4 M10K memory blocks.

Transformation coalescing has two layers of multiplexers, one for coalescing the coefficients vertically, and a second for coalescing horizontally as illustrated in Figure 18. Coalescing destroys the symmetry, so all 5 rows of the filter are required, requiring more flip-flops. For the FIR filter, this also requires more DSP blocks. The clock speed for the FIR filter is lowered significantly by the coalescing multiplexers; this is recovered with minimal additional resources by inserting pipeline registers between the multiplexers and multipliers (these registers were absorbed into the inputs of the DSP blocks). The clock speed for the morphological filter is barely affected, although the critical path does change from the row buffer operation to the coefficient coalescing. However, the additional delay is negligible, so pipelining the coefficient coalescing only serves to increase resources and has negligible effect on the maximum clock frequency.

Modifying the combination chain can share the common multiplier terms for the FIR filter and requires additional logic for the adders and multiplexers. Since the filter function is naturally pipelined in transpose form, there is no loss of speed. Rather than perform the division by 256 at the output (rescaling by 0 bits for the rows then 8 bits at the output (0/8)), two partial rescaling options were considered: rescaling by 4 bits from the row filters (4/4) or 6 bits (6/2). The 4/4 option reduces the row buffer word width to 12 bits (48 bits for the 4 buffers) which can be packed into 5 M10Ks. Dropping a further 2 bits (the 6/2 option) reduces the row buffers to 10 bits wide (40 bits total), requiring only 4 M10Ks. The small reduction in logic results from partial rescaling requiring smaller adders within the column adder chain; however, the associated reduction in memory is significant. The effect of distributed partial rescaling on the output image is insignificant (at most 1-pixel value).

For the morphological filter, the combination chain modification allows the offsets to be reused, giving a significant drop in logic resources required. Again, since the maximum combinations are pipelined, there is no effect on the operating frequency.

It is instructive to compare these designs with the same filters realized using the traditional direct form. The row buffers are on the input, requiring only 8 bits per buffer (32 bits total). This is the same as the transpose-form morphological filter but is less than the transpose-form FIR filter.

The 5 DSPs for the FIR filter come from the multiplication by coefficients 25 and 41; the others are optimized to additions. The multiplex network [11,15] adds significantly to the logic requirements of the FIR filter, but not so much on the morphological filter. This latter effect may be the synthesis optimizing the logic by combining the multiplexer and the offset operation. The overlapped prime and flush approach [10] has significantly fewer multiplexers, but this comes at the cost of additional priming registers. For the morphological filter, the increase in resources is from the multiplexer and offset operations no longer being combined. The operating speed is the same for these designs, and results from the combination chains; row adder chains (in parallel) feed into a column adder chain for the FIR filter, and similarly maximum combination chains for the morphological filter. Replacing these chains by a tree structure reduces the propagation delay by a factor of approximately 2 for the

FIR filter, and to a lesser extent for the morphological filter. The tree structure also enables the common multipliers and offsets to be factored out, reducing the number of DSP blocks and logic. This provides a more realistic comparison with the coefficient sharing used by the combination chain modification scheme. The trade-off between using the multiplexer network or the overlapped prime and flush comes down to the number of logic cells and registers required, so would only depend on resource availability. The direct-form implementation though is still significantly slower than the transpose form (running at 60% speed for the FIR filter and at 36% speed for the morphological filter), with both direct and transpose forms having similar resource requirements. The difference in speed can be addressed by pipelining the combination tree, although this comes at the expense of increasing the number of registers required (results for a 2-stage pipeline are shown in Table 1, if necessary further pipelining could be used).

*6.2. Experiment 2: Comparison of Border Management Schemes*

The second experiment was to compare the resources required for the different border management schemes. The results are presented in Table 2.

**Table 2.** Transpose-form resource requirements for different border management schemes (adaptive lookup tables (ALUT), flip-flops (FF), 10 kbit memory blocks (M10K), DSP blocks (DSP), maximum clock frequency (Fmax) in MHz).

| Method | 5 × 5 FIR Filter | | | | | 5 × 5 Morphological Filter | | | |
|---|---|---|---|---|---|---|---|---|---|
| | **ALUT** | **FF** | **M10K** | **DSP** | **Fmax** | **ALUT** | **FF** | **M10K** | **Fmax** |
| Zero-extension | | | | | | | | | |
| – Coalescing | 215 | 130 | 7 | 15 | 174 | 660 | 250 | 4 | 185 |
| – Combination chain | 323 | 239 | 7 | 2 | 185 | 366 | 168 | 4 | 184 |
| Constant extension | 387 | 130 | 7 | 15 | 173 | 636 | 250 | 4 | 175 |
| Duplication | 448 | 247 | 7 | 2 | 159 | 576 | 170 | 4 | 138 |
| Two-phase duplication | 389 | 248 | 7 | 2 | 188 | 436 | 170 | 4 | 171 |
| Mirroring | 410 | 251 | 7 | 2 | 180 | 439 | 171 | 4 | 172 |
| Mirroring with duplication | 445 | 247 | 7 | 2 | 178 | 540 | 170 | 4 | 172 |

The two forms of zero-extension were transformation coalescing or to modify the combination chain. For the FIR filter, coalescing consisted of setting the filter coefficients to 0 for pixels outside the image and using DSP units for the multiply and accumulate. For the morphological filter, the offsets for pixels outside the image were set to $-\infty$ (in practice $-255$ for 8-bit images). Combination chain modification enabled the transformations to be shared (reuse of multipliers for FIR, or offsets for the morphological filter). The increase in resources for the FIR filter reflect the movement of adders and registers from the DSP blocks into logic. However, for the morphological filter, sharing the offsets significantly reduces resources.

Constant extension was managed by multiplexing between the input pixel and the constant. Consequently, the pixel transformations could not be shared. For the FIR filter the multiplexers are the source of the 80% increase in logic resources. For the morphological filter, the multiplexers were combined with the offset calculation so that resources were similar to those for zero-extension.

The remaining extension methods were achieved by modifying the combination chain. The small differences in the number of flip-flops result from differences in the control chain. The logic resources reflect the complexity in terms of the number of adder/maximum units that could be reused, and complexity of the multiplexing by the difference schemes. Two-phase duplication has the lowest resource requirements because there are several combination terms that can be reused. Extension by duplication is slightly slower than the other methods because it requires multiple combination operations in two of the window positions, and this increases the propagation delay. For the morphological filter, two-phase duplication and mirroring without duplication require fewer

logic resources. It appears that for these two filters, the synthesis was able to combine the offsets with the combination chain giving better optimization. Otherwise there is little to distinguish between the different methods in terms of resources or speed.

In this experiment, for the FIR filters rescaling was performed at the end. With partial rescaling, it is expected that all these results would decrease by similar amounts to that shown in Table 1.

### 6.3. Experiment 3: Scalability with Window Size

For this experiment, a range of different morphological filters from $3 \times 3$ up to $15 \times 15$ using mirroring without duplication were synthesized to explore trends in resource requirements as a function of filter size. Combination chain modification was used because this is more efficient for implementing morphological filters. The results are listed in Table 3, with the resources normalized per pixel plotted in Figure 22.

**Table 3.** Resource requirements for the symmetrical morphological dilation filter as a function of filter size (adaptive lookup tables (ALUT), flip-flops (FF), 10 kbit memory blocks (M10K), maximum clock frequency (Fmax) in MHz). Normalized values are per window pixel. The last row shows the effect of pipelining.

| Filter Size | Raw Resource Count | | | | Normalized | |
|:---:|:---:|:---:|:---:|:---:|:---:|:---:|
| | ALUT | FF | M10K | Fmax | ALUT | FF |
| $3 \times 3$ | 163 | 93 | 2 | 196 | 18.1 | 10.3 |
| $5 \times 5$ | 439 | 171 | 4 | 172 | 17.6 | 6.8 |
| $7 \times 7$ | 924 | 279 | 5 | 174 | 18.9 | 5.7 |
| $9 \times 9$ | 1621 | 419 | 7 | 155 | 20.0 | 5.2 |
| $11 \times 11$ | 2539 | 591 | 8 | 140 | 21.0 | 4.9 |
| $13 \times 13$ | 3695 | 795 | 10 | 135 | 21.9 | 4.7 |
| $15 \times 15$ | 5090 | 1031 | 12 | 123 | 22.6 | 4.6 |
| $15 \times 15$ (pipelined) | 5054 | 2112 | 12 | 165 | 22.5 | 9.4 |

As expected, the resource requirements grow approximately in proportion to the number of pixels within the window. Part of this results from the filter function, which will be proportional to the window size, and part of this is for the combination network used to manage image borders. The number of ALUTs and FFs normalized by the number of window pixels makes this particularly clear. The number of ALUTs per window pixel increases because as the window grows larger, the width of the multiplexers in the combination network also grows. This growth is relatively slow, which shows that the combination network scales well with filter size. The higher numbers for the $3 \times 3$ window reflect the overhead of the control logic, which is a greater proportion for the small window. The number of FFs per window pixel decreases, reflecting the decreasing proportion used by the control logic and row buffer output (which are proportional to the window width rather than area). The asymptote here will be 4 FFs per window pixel reflecting the fact that only half of the row filters are required because of symmetry.

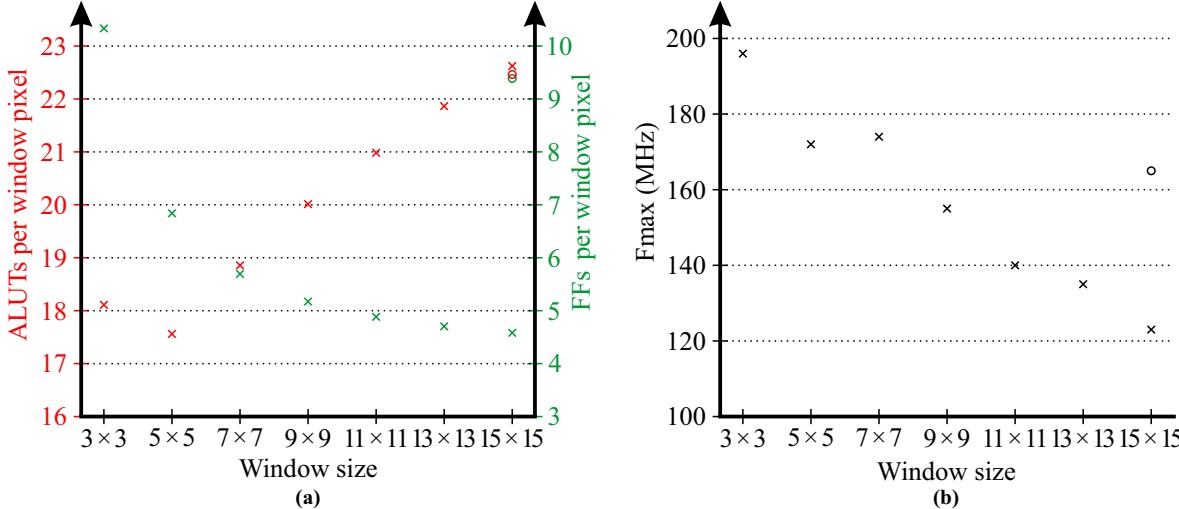

**Figure 22.** (**a**): resources required per window pixel; (**b**): maximum clock frequency. Pipelined results for the 15 × 15 window are shown with ○.

The maximum clock frequency decreases with increasing window size. This results purely from the increase in the complexity of the multiplexers within the combination network, as these form the critical path for windows larger than about 7 × 7. If clock speed is critical, this can be mitigated by pipelining the output of the combination networks, as shown by the last row in Table 3. Although this more than doubles the number of flip-flops required, in an FPGA implementation, these flip-flops are associated with the logic cells used to realize the combination network and would be otherwise unused. The critical path in this case is still the combination network, so the speed would still be expected to decrease for larger filters unless additional pipelining was used.

Overall, these results indicate that the transpose-form filters scale well with increasing window size.

## 7. Conclusions

This paper has demonstrated that border management using transpose filter structures in not only feasible, but can have significant advantages over direct-form filter structures.

For FIR filters, coefficient coalescing enables DSP blocks to be used for realizing the multiply, add, and associated register resulting in the fewest required resources. However, this comes at the cost of not being able to exploit multiplier reuse and symmetry. Modifying the combination chain separates the multiplication and addition, approximately doubling the resources by forcing the additions and registers into the FPGA fabric. Reflecting this trade-off, coefficient coalescing would therefore be recommended for implementing FIR filters on FPGAs with plentiful DSP blocks.

For FIR filters, using the transpose filter structures requires wider memories for the row buffers because these come between the multiplications and the distributed additions. Since scaling of the output occurs at the end, more bits must be carried through the row buffers. It is demonstrated that this increase in RAM resources by the transpose structure for FIR filters can be effectively managed through distributing the rescaling between the row filters (before the row buffers) and the output.

For morphological filters, transform coalescing means that symmetry cannot effectively be exploited, making this a less viable option than modifying the combination chain.

Overall, the resources required by the transpose form with combination chain modification are similar for both linear and morphological filters to that for direct-form filters using the multiplexer network, especially if distributed partial rescaling is used with FIR filters to reduce the width of the row buffers. The transpose-form filter structures scale well with increasing filter size, with resources only growing slightly faster than the number of pixels within the window. The key advantage of the

transpose form is that the filters are significantly faster than direct-form filters, primarily as a result of the pipelining inherent within their structure. Although direct-form filters can be pipelined to improve the speed, this comes at the cost of additional resources.

The disadvantage of using the transpose form for filter realization is that the filter function is no longer independent of the window formation. However, this paper has demonstrated that the complexities of border handling can be confined to an additional processing layer which implements the combination network. (For 2D filters, two such layers are required: one for the row filters, and one for the column combination.)

In conclusion, this paper has demonstrated that image border processing can effectively and efficiently be integrated within transpose-form filters, and that the complexity of the combination network is similar to that of the more conventional direct-form processing.

**Author Contributions:** Conceptualization, D.G.B.; Investigation, D.G.B. and A.S.A.; Methodology, D.G.B.; Project administration, D.G.B.; Supervision, D.G.B.; Validation, D.G.B. and A.S.A.; Writing—original draft, D.G.B.; Writing—review & editing, D.G.B. and A.S.A.

**Funding:** This research received no external funding.

**Conflicts of Interest:** The authors declare no conflict of interest.

## Abbreviations

The following abbreviations are used in this manuscript:

| | |
|------|-------------------------------|
| ALUT | Adaptive Lookup Table |
| DSP | Digital Signal Processing |
| FF | Flip-Flop |
| FIR | Finite Impulse Response filter |
| FPGA | Field Programmable Gate Array |
| LUT | Lookup Table |
| M10K | 10 kbit memory block |
| RTL | Register Transfer Level |
| SRL | Shift Register Logic |

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
