# Peer review of "Border Handling for 2D Transpose Filter Structures on an FPGA"

_2313-433X, doi:10.3390/jimaging4120138_

Reviewer 1 Report

The paper deals with an FPGA implemntation of transpose filter structure, with signal boundary handling for 2D image. The paper has a weak contribution in mind, but is well written and methodology is well explained and well designed.

The main remark that I do is that the authors should not only discuss about technical details and implementation, but also discuss about the usability and the need of such a preprocessing in the introduction. It is not obvious that we need to extrapolate pixels to handle border effects. If data is missing, filter could not be applicable, and applying a filter on extrapolated data is source of errors in applications. What are the issues that arise when image size is reduced? Moreover, what is the usability of using these data? In other words, what are the issues when border handled data and extrapolated are used in computee vision or machine vision algorithms? It is not discussed at all in the introduction. The authors should add references in the introduction to explain why it is important to do extrapolation when filtering images.

Author Response

Point 1: The main remark that I do have is that the authors should not only discuss about technical details and implementation, but also discuss about the usability and the need of such a preprocessing in the introduction. It is not obvious that we need to extrapolate pixels to handle border effects. If data is missing, filter could not be applicable, and applying a filter on extrapolated data is source of errors in applications. What are the issues that arise when image size is reduced? Moreover, what is the usability of using these data? In other words, what are the issues when border handled data and extrapolated are used in computer vision or machine vision algorithms? It is not discussed at all in the introduction. The authors should add references in the introduction to explain why it is important to do extrapolation when filtering images. Author's response: Thank you for pointing out this oversight in our introduction. We have expanded the section in the introduction on border handling to discuss some of the issues raised - when border handling is unnecessary, some example applications which benefit from appropriate extrapolation, and an outline of the potential dangers of extrapolation.

Reviewer 2 Report

Dear Authors,

the paper gives comprehensive introduction and clearly states the problem, presents the solutions and shows verification in the FPGA. It is well written, and the pictures' quality is high. Before I can recommend it for publication, please consider the following remarks:

- Fig. 1 can be improved in terms of placement of window rather at the edge. Also the input pixel stream can be represented rather by a meander arrow to make it more meaningful. Meander can go back between the pixels (or use dash arrow during return). In my opinion it will be more readable.

- Even though normalized values are presented (Fig. 3), it would be nice to have also graph showing the relation between selected parameters (e.g. utilization of selected resources and maximum speed) versus window size.

Author Response

Point 1:

Fig. 1 can be improved in terms of placement of window rather at the edge. Also the input pixel stream can be represented rather by a meander arrow to make it more meaningful. Meander can go back between the pixels (or use dash arrow during return). In my opinion it will be more readable.

Author's response:

Thank you for the suggestions. The window in Figure 1 has been placed in the top left corner, and a dotted link has been added between the lines to more clearly show the raster streaming.

Point 2:

Even though normalized values are presented (Table 3), it would be nice to have also graph showing the relation between selected parameters (e.g. utilization of selected resources and maximum speed) versus window size.

Author's response:

In addition to Table 3, Figure 22 has been added to graphically show the normalised resources and maximum frequency versus window size.

Round  2

Reviewer 1 Report

The authors responded to my main question.